# Demonstrative Experiment on the Favorable Effects of Static Electric Field Treatment on Vitamin D_3_-Induced Hypercalcemia

**DOI:** 10.3390/biology10111116

**Published:** 2021-10-29

**Authors:** Tohru Kimura, Kengo Inaka, Noboru Ogiso

**Affiliations:** 1Laboratory Animal Science, Joint Faculty of Veterinary Medicine, Yamaguchi University, Yamaguchi City 753-8515, Japan; b001tn@yamaguchi-u.ac.jp; 2National Center for Geratrics Gerontology, National Institute for Longevity Sciences, Obu City 474-8511, Japan; ogiso@ncgg.go.jp

**Keywords:** calcification, cardiac muscles, complementary and integrative medicine, electrolytes, gastric mucosa, histopathology, renal tissues, serum biochemistry, static electric field treatment, vitamin D_3_-induced hypercalcemia

## Abstract

**Simple Summary:**

Static electric field (SEF) treatment by high-voltage alternating current is a traditional complementary medicine in Japan. Although it is believed that the SEF-induced electric current serves to regulate cellular or humoral responses in patients, the mechanism for SEF treatment remains poorly understood. There have been very few experimental reports on the biological action with SEF treatment. The aim of this study was to elucidate the effects of SEF treatment on vitamin D_3_ (Vit D_3_)-induced abnormalities in mice. SEF treatment improved the abnormalities in the renal function tests and the imbalance of serum electrolytes. In addition, this treatment remarkably attenuated the Vit D_3_-induced tissue injuries (severe tissue calcification in the kidneys, hearts, and stomachs). It was likely that the SEF treatment had some favorable effects on the metabolism of calcium. In conclusion, this study provides important evidence that SEF treatment can reduce hypercalcemia and remove calcium deposits from the renal, cardiac, and gastric tissues. SEF treatment is useful in the regulation of disorders caused by an imbalance of serum electrolytes. This study experimentally demonstrates the favorable effects of SEF treatment on Vit D_3_-induced hypercalcemia. For small animals, the larger the body surface area per body weight becomes, the higher the therapeutic efficacy with SEF treatment.

**Abstract:**

The purpose of this study was to elucidate the effects of static electric field (SEF) treatment on vitamin D_3_ (Vit D_3_)-induced hypercalcemia and renal calcification in mice. The mice were assigned to three groups: Vit D_3_-treated mice, mice treated with Vit D_3_ and SEF (Vit D_3_ + SEF), and untreated mice. After the administration of Vit D_3_, the Vit D_3_ + SEF-treated mice were exposed to SEF treatment by a high-voltage alternating current over five days. Serum biochemical examinations revealed that both the creatinine and blood urea nitrogen concentrations were significantly higher in the Vit D_3_-treated group. Significantly, decreased Cl concentrations, and increased Ca and inorganic phosphorus concentrations, were found in the Vit D_3_-treated group. In the Vit D_3_ + SEF-treated group, these parameters returned to the levels of the untreated group. In the Vit D_3_-treated group, histopathological examinations showed marked multifocal calcification in the lumens of the renal tubules and the renal parenchyma. The myocardium was replaced by abundant granular mineralization (calcification), with degeneration and necrosis of the calcified fibers. The stomach showed calcification of the cardiac mucosa. SEF treatment remarkably attenuated the Vit D_3_-induced hypervitaminotic injuries. In conclusion, this study provides important evidence that SEF treatment can reduce hypercalcemia and remove calcium deposits from the renal, cardiac, and gastric tissues. SEF treatment is useful in the regulation of disorders caused by an imbalance of serum electrolytes.

## 1. Introduction

Static electric field (SEF) treatment by high-voltage alternating current is a traditional complementary medicine used for headache, shoulder stiffness, chronic constipation, and insomnia. The widespread use of static electric field treatment has been accepted as a self-care physical intervention for pain reduction [1]. The safety of long-term continuous exposure has been confirmed by a study using mice [2]. The effect of the alternating current electrostatic high-voltage potential load was reported by a Japanese investigator in 1961 [3]. SEF exposure induced changes in the serum electrolytes (Ca, Mg, and phosphorus) of rabbits. It is believed that the SEF-induced electric current serves to regulate the cellular or humoral responses in patients [4]. There have been very few experimental reports on the biological action with SEF treatment (one study reports the inhibition of a rheumatoid arthritis model in mice) [5].

It is well known that hypervitaminosis D_3_ causes an imbalance of the electrolyte metabolism characterized by hypercalcemia. The presence of microcalcification in the soft tissues is pathognomonic of the microscopic finding [6]. Although the exact mechanism of vitamin D toxicity remains unclear, recent studies report that excessive vitamin D supplementation caused critical clinical signs, such as myocardial infarction, altered mental status, or neuromyelitis spectrum disorder in elderly patients, as well as myositis, rhabdomyolysis, and severe hypercalcemia in a body builder [7,8]. In addition, the clinical presentation of an infant patient who was vulnerable to vitamin D toxicity included irritability, anorexia or poor feeding, diarrhea, the failure to gain weight, constipation, nausea, vomiting, dehydration, and physiological manifestations (hypercalcemia, hyperphosphatemia, and nephrocalcinosis) [9].

In the present study, we used this experimental model to investigate the regulation of the electrolyte imbalance after SEF treatment. The purpose of this study was to elucidate the effects of SEF treatment on vitamin D_3_ (Vit D_3_)-induced hypercalcemia and renal calcification in mice. In addition, we biochemically and histopathologically investigated the sera and found that SEF treatment was useful in the field of complementary and integrative medicine.

## 2. Materials and Methods

### 2.1. Animals

Fifteen female six-week-old ICR mice, weighing 24–27 g, were purchased from Kyudo Co., Ltd. (Tosu, Saga, Japan).

The mice were housed in polysulfone cages (W 184 × D 332 × H 147 mm, Clea Japan Inc., Tokyo, Japan), with bedding made from pure-pulp (Care-feeaz, Hamri Co., LTD., Tsukuba, Ibaraki, Japan), in our specific-pathogen-free (SPF) barrier facilities. The animal room was maintained at constant temperature (23 ± 1 °C) and relative humidity (55 ± 10%). The room air was ventilated 10 to 15 times per hour automatically, and a 12 h/12 h light–dark cycle (lighting 06:00–18:00) was imposed. The animals received commercial radiation-sterilized diets (CE-2, Clea Japan Inc., Tokyo, Japan) and water ad libitum. These diets were previously sterilized by γ-irradiation at a dose of 30 kGy. The health statuses of the animals in our SPF facilities were checked four times a year for the pathogens listed by the Japanese Association of Laboratory Animal Facilities of National University Corporations, and the animals were free from all these pathogens.

All procedures involving animals were approved by the Institutional Animal Care and Use Committee of Yamaguchi University and followed the Guidelines of Animal Care and Experiments of Yamaguchi University. The animal care and use program for the Advanced Research Center for Laboratory Animal Science in Yamaguchi University has been accredited by AAALAC International since 2018. The Institutional Animal Care and Use Committee of Yamaguchi University approved this specific study under Approval No. 368.

### 2.2. Procedures

Mice were assigned randomly to the following three groups: 1. Vit D_3_-treated mice (*n* = 5); 2. Vit D_3_ and SEF (Vit D_3_ + SEF)-treated mice (*n* = 5); and 3. Untreated mice (*n* = 5).

The mice (Vit D_3_ and Vit D_3_ + SEF-treated mice) were administrated 2.5 μg/kg (100 IU/kg) of activated Vit D_3_ (calcitriol, Rocalcitrol injection, Kyowa Hakko Kirin Co., Ltd., Tokyo, Japan), intraperitoneally for three consecutive days. The doses of Vit D_3_ were chosen based on the following results: 1. There were no significant effects in mice given doses of 5–250 μg/kg; 2. After 10 days of treatment with 2.5 μg/kg/day, calcification was observed in the kidneys in rats [10]; 3. In mice given doses (0.00125, 0.0125, 0.0625, 0.125, 0.25, 0.5, and 2.0 μg/kg), the dose of Vit D_3_, leaving all mice alive for five weeks, was up to 0.25 μg/kg [11].

After completion of the administration of Vit D_3_, the mice were exposed to an SEF treatment of 60 Hz at 110 mT intensity by a high-voltage alternating current (1414 V to 9899 V) for 60 min/day. This treatment was performed for five consecutive days.

In this study, we used a high-voltage potential treatment device (Live-max 1/f for veterinary medicine, COCOROCA Corporation, Tokyo, Japan), which was approved by the Ministry of Health, Labor and Welfare (Approval Number 227AIBZX00017000). The cages containing the mice were placed on an electrical potential loading mat (304 × 420 mm, COCOROCA Corporation, Tokyo, Japan), and then the cages were exposed to static electric field treatment. All of the procedures were conducted on a high-voltage insulating mat (800 × 1200 mm, COCOROCA Corporation, Tokyo, Japan).

### 2.3. Serum Biochemistry

Under systemic anesthesia with isoflurane, the blood samples were collected from the caudal vena cava of the mice, using no anticoagulant. At 30 min after collection of the blood samples, sera were separated by centrifugation at 1500× *g* for 10 min for biochemical analysis.

The following parameters were measured using a blood chemistry analyzer (Dry Chem NX 500 V: Fuji Film Co. Ltd., Tokyo, Japan): 1. Renal function tests (total protein (TP), albumin (ALB), albumin: globulin (A/G) ratio, blood urea nitrogen (BUN), and creatinine (CRE)); 2. Hepatic function tests (aspartate aminotransferase (AST), alanine aminotransferase (ALT), and alkaline phosphatase (ALP)); and 3. Electrolytes (Na, K, Cl, Ca, inorganic phosphorus (IP), and Mg).

### 2.4. Histopathology

Tissue specimens were obtained from the main organs (lungs, hearts, livers, and kidneys). The tissue specimens were fixed in 10% neutral buffered formalin, and 4 μm sections were stained with hematoxylin and eosin (HE), and by von Kossa stain.

### 2.5. Statistical Evaluation

Values were expressed as the mean ± standard deviation (SD), and statistical analysis was performed using the one-way repeated measures analysis of variance (ANOVA), and a multiple comparison test for parametric data (Tukey–Kramer method) and for nonparametric data (Steel’s method). A comparison review of the aforementioned parameters was carried out among the three groups, and statistical significance was established at *p* < 0.05 or *p* < 0.01. Statistical analyses were performed with statistic software (Statcel-the Useful Add in Forms on Excel, 4th ed. OMS Publication Ltd., Tokorozawa, Saitama, Japan).

## 3. Results

### 3.1. Serum Biochemical Findings

The results of the TP and ALB concentrations and the A/G ratios are shown in Figure 1. Although there were, statistically, no significant differences in the TP and ALB concentrations among the three groups. In the Vit D_3_-treated group, declined ALB concentrations brought a relative decrease in A/G ratios, leaving the TP levels within the unaffected range (*p* < 0.01).

The changes in the renal function tests are shown in Figure 2. The CRE concentrations in the Vit D_3_-treated group were significantly higher than those in the Vit D_3_ + SEF-treated group and the untreated group (*p* < 0.05). There were no differences in the CRE concentrations between the Vit D_3_ + SEF-treated group and the untreated group. The BUN concentrations in the Vit D_3_-treated group were also significantly higher than those in the Vit D_3_ + SEF-treated group and the untreated group (*p* < 0.01). The BUN concentrations did not differ between the Vit D_3_ + SEF-treated group and the untreated group.

The changes in the electrolytes are shown in Figure 3. The Na, K, and Mg measurements exhibited no differences among the three groups. In contrast, significant decreases in the Cl levels in the Vit D_3_-treated group were statistically found, as compared with those in the Vit D_3_ + SEF-treated group and the untreated group (*p* < 0.01). There were no differences in the Cl levels between the Vit D_3_ + SEF-treated group and the untreated group. The Ca levels in the Vit D_3_-treated group indicated significantly higher increases than those in the Vit D_3_ + SEF-treated group and the untreated group (*p* < 0.05 and *p* < 0.01, respectively). There were no differences in the Ca levels between the Vit D_3_ + SEF-treated group and the untreated group. The IP levels in the Vit D_3_-treated group were significantly higher than those in the untreated group (*p* < 0.05). There were no differences in the IP levels between the Vit D_3_ + SEF-treated group and the untreated group.

There were statistically no significant differences in the hepatic function enzymes (AST, ALT, and ALP) among the three groups (Figure 4). Although the AST and ALT activities tended to increase in the mice treated with Vit D_3_, the Vit D3-treated group did not show significant differences as compared with the Vit D_3_ + SEF-treated group and the untreated group.

### 3.2. Histopathological Findings

In the Vit D_3_-treated group, the gross findings revealed that the kidneys were bilaterally pale with irregular, granular, and white spotted surfaces. The renal specimens showed shrinkage of the vascular glomeruli, tightness of the glomerular space, and the degeneration of the epithelial lining of the renal tubules. Histopathologically, coagulation necrosis of the renal tubular epithelium was found, along with degenerative glomerulus. Marked multifocal calcification was observed in the lumens of the renal tubules and renal parenchyma (Figure 5A, stained area ratio: 25–40% in 10 tissue sections). The tubular basement membranes contained calcified deposits, and the tubular epithelium was sloughed within the lumen. The calcification disappeared from the renal tissues in the Vit D_3_ + SEF-treated group (Figure 5B).

Three mice showed some whitish spots on the cardiac surfaces. Myocardium was replaced by abundant granular mineralization (calcification), with degeneration and necrosis of the calcified fibers (Figure 6, stained area ratio: 15–20% in 10 tissue sections). The foci had fibrosis with moderate lymphocytic infiltration and mild calcification of the blood vessels. The stomach showed calcification of the muscularis externa, muscularis mucosa, and mucosa, admixed with vessel calcification (Figure 7, stained area ratio: 10–15% in 10 tissue sections).

In the Vit D_3_ + SEF-treated group, this treatment remarkably attenuated the Vit D_3_-induced hypervitaminotic injuries. Calcification, in particular, in the renal tubules, apparently decreased, as compared with that observed in the Vit D_3_-treated group. The glomeruli were unaffected, and the multifocal mineralized deposits resolved with SEF treatment in the the tubular lumen and basement membranes. No apparent changes were found in the hearts and livers, indicating the restoration of tissue damage by SEF treatment.

The untreated group showed no histopathological changes in any of the tissues, including the renal structure.

## 4. Discussion

SEF treatment, the placing of patients in the SEF by a high-voltage alternating current, has been utilized as one of the effective physical therapies. Few reports are available to evaluate the clinicopathological changes in the living body following SEF treatment [12,13,14,15]. Other investigators have described that extremely low-frequency electric fields had an impact on the lipid metabolism parameters (significant decreases in triglycerides and free fatty acids) in a hind-limb ischemic rat [4]. In addition, exposure to static magnetic fields had time-dependent effects on the glucose and lipid metabolisms in rats [16,17].

In the mice used in this study, the changes in the electrolytes, and the calcium deposition in the soft tissues due to Vit D_3_ toxicity, were in good agreement with those of the other reported investigations [18,19]. Vit D_3_ treatment caused hypercalcemia and hyperphosphatemia to a certain degree.

Previous studies have reported that extremely low-frequency electric fields had an impact on cellular Ca^2+^ regulation in mouse splenocytes and human vascular endothelial cells [12,20]. Several studies have found that static magnetic fields influenced cellular calcium homeostasis [21,22]. The present study demonstrates that SEF treatment has the ability to participate in the regulation of Vit D_3_-induced hypercalcemia. Exposure to a high-voltage alternating current histologically inhibits the development of experimental collagen-induced arthritis in mice [5]. It was likely that SEF treatment had some favorable effects on the metabolism of calcium. The significant reduction in the Cl and IP concentrations was also accompanied by a decrease in the Ca concentrations following SEF treatment. Iatrogenic vitamin D toxicity in an infant was managed during hospital stay by the administration of fluids, calcitonin, and bisphosphonates. After therapy, the infant showed normal levels of Ca and PI, and there was no specific intervention to reverse the nephorocalcinosis, despite the good long-term renal function prognosis [9]. Our results suggest that SEF treatment should provide relief of tissue calcification, resulting in the normalization of electrolytes, such as Ca, PI, and Cl.

According to the experimental results of the spinal cords of rats, the Mg^+^ concentrations were unchanged by acute exposure to static magnetic fields [23]. Our results of the serum Mg^+^ concentrations were very different from those stated in the above-described report. This difference was probably dependent on the characteristics of the specimens (spinal fluid and sera) taken from the laboratory animals.

A recent review described that exposure to static magnetic fields altered the plasma levels of vitamins A, C, D, and E. Exposure to SEF probably had a direct effect on the metabolism of Vit D_3_, leading to the remarkable changes in the electrolytes [17].

The Vit D_3_-treated mice showed significant increases in the CRE and BUN values and the A/G ratios, suggesting decreases in the renal function. The CRE and BUN values and the A/G ratios in the Vit D_3_ + SEF-treated mice were approximately equal to those obtained from the untreated normal mice. These findings reveal that SEF treatment restored the renal function to relatively normal limits.

Although there was a variation in measurements, depending on individual differences, Vit D_3_ treatment caused an elevation in AST and ALT activities. SEF treatment provided the restoration of these elevated parameters to the state in the untreated mice, and the hepatic function consequently returned to normal.

Kidneys have naturally regenerative potentialities and can fully recover after acute injury under favorable conditions [24]. Our histopathological examinations demonstrated that SEF treatment brought remarkable improvements to the renal tubular epithelial cells and the glomerular membranes suffering from acute renal calcification. The calcium deposits were prominently removed from the renal tubules. These histopathological findings accurately reflected the changes in the serum chemical profiles, particularly in the electrolyte and renal functional examinations. Electric fields affect the selective transport of ions or molecules through membranes [25,26,27,28]. A previous experiment reported that the calcium influx was elevated 1.5-fold when the lymphocytes were exposed to Con-A plus 60 Hz magnetic fields [26]. In calcium transport and binding, the cell surface is implicated as a major site of interaction for extremely-low-frequency electromagnetic fields [29,30,31]. For small animals, the larger the body surface area per body weight becomes, the higher the therapeutic efficacy with SEF.

In this examination, although echocardiograms could not be recorded in the mice, the Vit D_3_-treated group seemed to be slow in their behavior, suggesting the changes in the cardiac legions and function.

In the Vit D_3_ + SEF-treated mice, there were no histopathological changes in the other tissues, including the hearts and stomachs. These histopathological results were in close accord with the aforementioned serum chemical profiles, such as the enzymatic activities and the electrolyte levels. SEF treatment resulted in the elimination of Vit D_3_-induced invasiveness (calcification).

## 5. Conclusions

In conclusion, on the basis of the biochemical and histopathological investigation of sera, this study provides important evidence that SEF treatment can reduce hypercalcemia and remove calcium deposits from renal tissues, resulting in the restoration of Vit D_3_-induced renal injury. SEF treatment is useful in the regulation of disorders caused by an imbalance of serum electrolytes.

## Figures and Tables

**Figure 1 biology-10-01116-f001:**
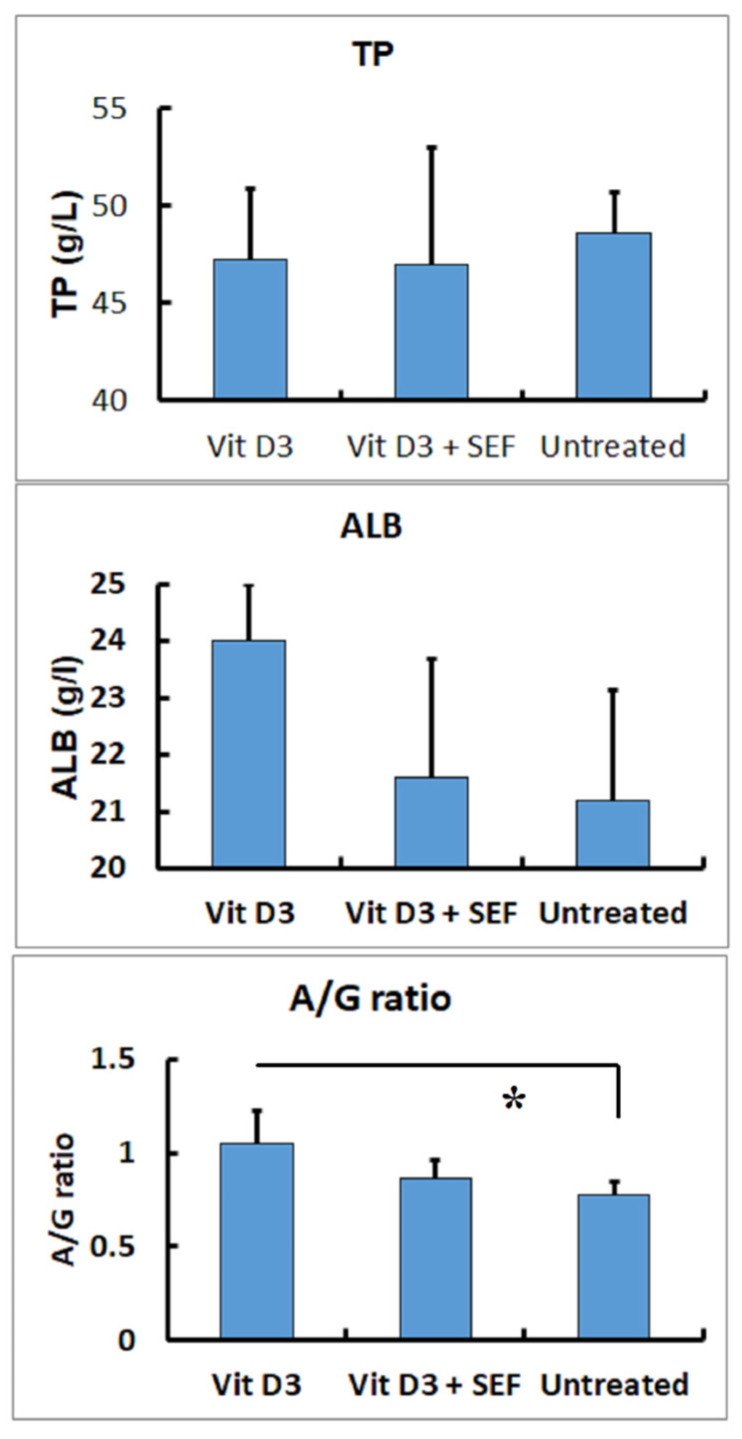
Serum biochemical findings in TP and ALB concentrations and A/G ratios. No significant differences are seen in TP and ALB concentrations among the 3 groups. The Vit D_3_-treated group shows a decrease in A/G ratios. *: Significantly different between the Vit D3-treated and the untreated groups (*p* < 0.05).

**Figure 2 biology-10-01116-f002:**
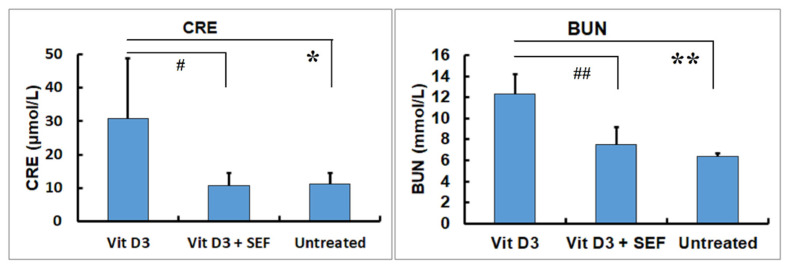
Serum biochemical findings in CRE and BUN concentrations. Both CRE and BUN concentrations in the Vit D_3_-treated group increase. *: Significantly different from the Vit D_3_ + SEF-treated group and the untreated group (*p* < 0.05). **: Significantly different from the Vit D_3_ + SEF-treated group and the untreated group (*p* < 0.01). ^#^: Significantly different from the Vit D_3_ + SEF-treated group and the untreated group (*p* < 0.05). ^##^: Significantly different from the Vit D_3_ + SEF-treated group and the untreated group (*p* < 0.01).

**Figure 3 biology-10-01116-f003:**
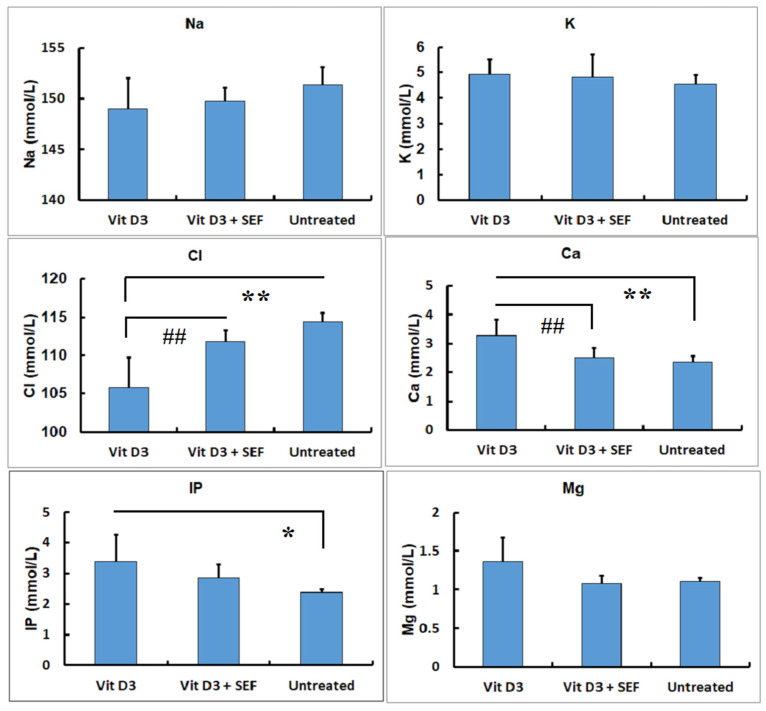
Serum biochemical findings in electrolytes (Na, K, Cl, Ca, inorganic phosphorus (IP), and Mg). Decreased Cl concentrations and increased Ca and IP concentrations are seen in the Vit D_3_-treated group. *: Significantly different from the Vit D_3_ + SEF-treated group and the untreated group (*p* < 0.05). **: Significantly different from the Vit D_3_ + SE-treated group and the untreated group (*p* < 0.01). ^##^: Significantly different from the Vit D_3_ + SEF-treated group and the untreated group (*p* < 0.01).

**Figure 4 biology-10-01116-f004:**
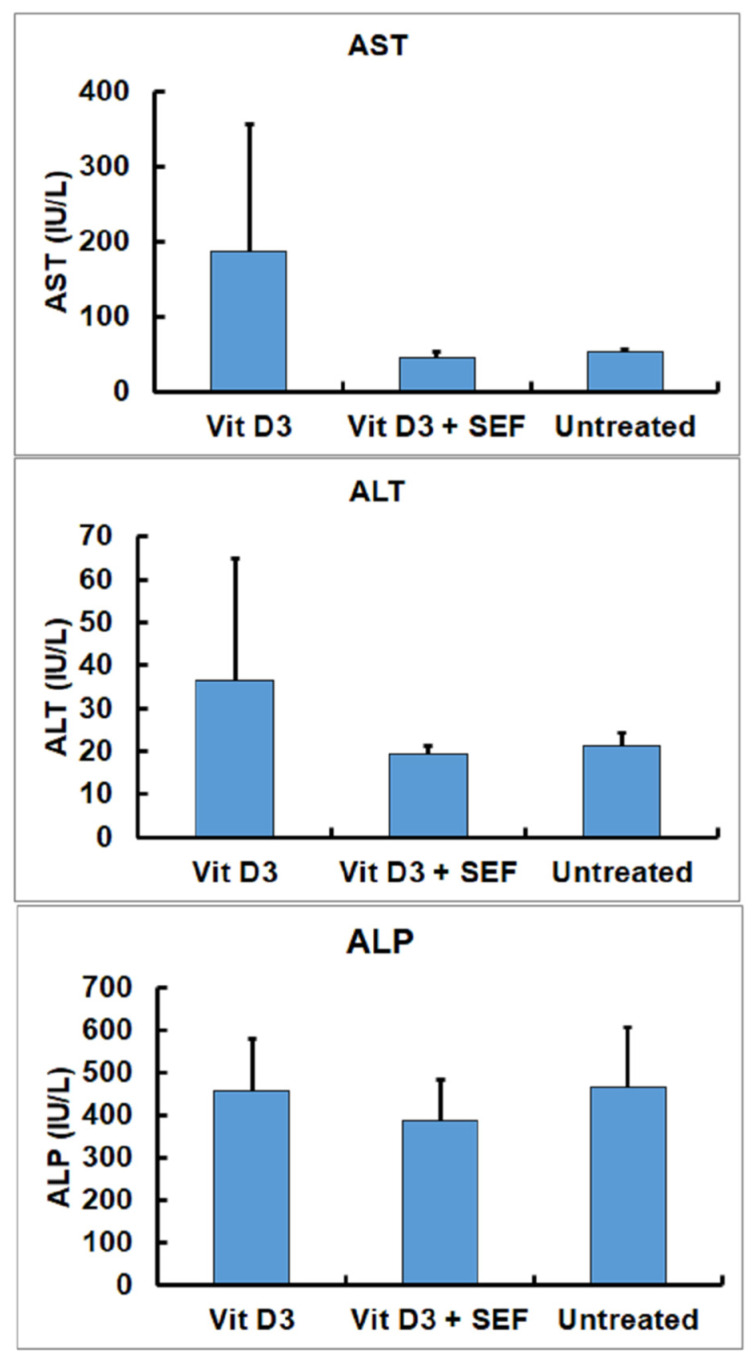
Serum biochemical findings in hepatic functional enzymes (AST, ALT, and ALP). AST and ALT activities tended to increase in the Vit D_3_-treated group.

**Figure 5 biology-10-01116-f005:**
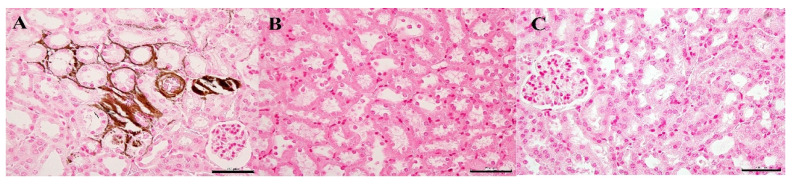
Histopathological changes in the renal tubules. (**A**) Vit D_3_-treated mice. Marked multifocal calcification is observed in the lumens of renal tubules and renal parenchyma. von Kossa stain. Bar = 50 μm. (**B**) Vit D_3_ + SEF-treated mice. Calcification is not found in the renal tissues. von Kossa stain. Bar = 50 μm. (**C**) Untreated mice. von Kossa stain. Bar = 50 μm.

**Figure 6 biology-10-01116-f006:**
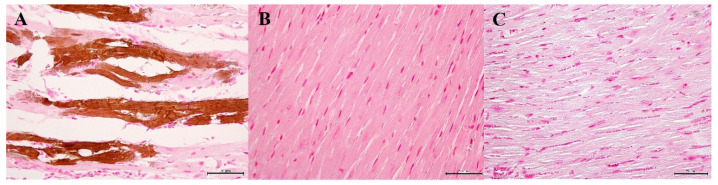
Histopathological changes in the cardiac muscles. (**A**) Vit D_3_-treated mice. Abundant granular mineralization (calcification) with degeneration and necrosis of the calcified fibers is noted in the cardiac muscles. von Kossa stain. Bar = 50 μm. (**B**) Vit D_3_ + SEF-treated mice. Calcification is not found. von Kossa stain. Bar = 50 μm. (**C**) Untreated mice. von Kossa stain. Bar = 50 μm.

**Figure 7 biology-10-01116-f007:**
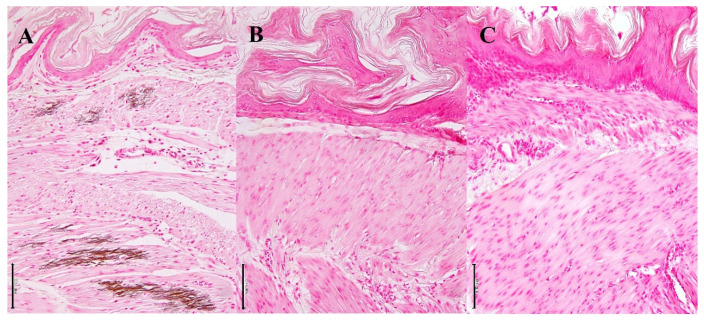
Histopathological changes in the gastric mucosa. (**A**) Vit D_3_-treated mice. Calcification is seen in the gastric mucosa. von Kossa stain. Bar = 100 μm. (**B**) Vit D_3_ + SEF-treated mice. Calcification is not found. von Kossa stain. Bar = 100 μm. (**C**) Untreated mice. von Kossa stain. Bar = 100 μm.

## Data Availability

Data are available upon reasonable request to the corresponding author.

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
