# Peer review of "Demonstrative Experiment on the Favorable Effects of Static Electric Field Treatment on Vitamin D3-Induced Hypercalcemia"

_biology, 2021, doi:10.3390/biology10111116_

Round 1

Reviewer 1 Report

The authors investigated the significance of static electric field treatment focusing on Vitamin D3 induced abnormalities in mice. Authors’ results are remarkable to identify the changes of tissue calcifications due to static electric field treatment. However, the reviewer has some concerns about the explanation of introduction, results, and discussion. Please consider and reply for comments and questions.

Major

  1. Please describe the impact of calcification to our health in introduction. This contributes to the significance of authors’ manuscript.
  2. Authors should show representative figures in untreated group. In addition, authors should the quantitative analyses like stained area ratio because the reviewer suspected that those changes (calcifications) can affect liver or renal function.
  3. It is better that authors show the changes of cardiac function among three groups if authors have echocardiogram data because authors didn’t show the clinical effect on cardiac tissue calcification.
  4. Did authors check urine protein? According to authors’ discussion, the change of A/G ratio was affected by impaired renal function, but it was also affected by impaired liver function. If authors can show at least urine protein, it can support authors’ discussion.
  5. Please explain deeply why Cl and IP reduced significantly accompanied by a decrease in Ca concentration following static electric field treatment.

Minor

  1. Figure 4 was described as Figure 1, and AST/ALT were mentioned in line 134.
  2. Please mention the institutional animal care and use committee of Yamaguchi University (line 123-129) to 1 Animals.
  3. Authors mentioned “Lungs” in line 216 and 272. Please show the figures of lungs or delete this word.

Author Response

The authors investigated the significance of static electric field treatment focusing on Vitamin D3 induced abnormalities in mice. Authors’ results are remarkable to identify the changes of tissue calcifications due to static electric field treatment. However, the reviewer has some concerns about the explanation of introduction, results, and discussion. Please consider and reply for comments and questions.

I am grateful to review #1 for the critical comments and useful suggestions that have helped me to improve this paper. As indicated in the response that follow, I have taken all these comments and suggestions into account in the revised version of this paper.

Major

  1. Please describe the impact of calcification to our health in introduction. This contributes to the significance of authors’ manuscript.

The sentence has been added as suggested.

Page 2, Lines 60-64

“Although the exact mechanism of Vitamin D toxicity remains unclear, recent studies described that excessive Vitamin D supplementation caused critical clinical signs such as myocardial infarction, altered mental status or neuromyelitis spectrum disorder in elderly patients and myositis, rhabdomyolysis and severe hypercalcemia in a body builder [7, 8].”

  1. Authors should show representative figures in untreated group. In addition, authors should the quantitative analyses like stained area ratio because the reviewer suspected that those changes (calcifications) can affect liver or renal function.

We have added new microscopic photographs (C. Untreated group) in Figures 5, 6 and 7.

We showed the quantitative analyses like stained area ratio.

Page 6, Lines 197+198

(Figure 5-A, stained area ratio: 25-40% in 10 tissue sections)

Page 7, Lines 209, 212-213

(Figure 6, stained area ratio: 15-20% in 10 tissue sections)

(Figure 7, stained area ratio: 10-15% in 10 tissue sections).

  1. It is better that authors show the changes of cardiac function among three groups if authors have echocardiogram data because authors didn’t show the clinical effect on cardiac tissue calcification.

We could not unfortunately record echocardiograms in the mice.

Page 9, Lines 288-290

We described the sentence in Discussion. 

“In this examination, although echocardiograms could not be recorded in the mice, the Vit D3 treated group seemed to be slow in their behavior, suggesting the changes in cardiac legions and function.”

  1. Did authors check urine protein? According to authors’ discussion, the change of A/G ratio was affected by impaired renal function, but it was also affected by impaired liver function. If authors can show at least urine protein, it can support authors’ discussion.

We could not perform the urine analysis including urine protein in this study.

We want to try the urine examination in the next study.

  1. Please explain deeply why Cl and IP reduced significantly accompanied by a decrease in Ca concentration following static electric field treatment.

Page 8, Lines 251-257

The sentences have been added as suggested.

“Iatrogenic Vitamin D toxicity in an infant was managed during hospital stay by fluids and calcitonin and bisphosphonates administration. After therapy, the infant showed the normal levels of Ca and PI and there was no specific intervention to reverse the nephorocalcinosis despite the good long-term renal function prognosis [9]. Our results suggested that SEF treatment should provide relief of tissue calcification, resulting in normalization in electrolytes such as Ca, PI and Cl.”

Minor

  1. Figure 4 was described as Figure 1, and AST/ALT were mentioned in line 134.

We corrected our mistakes.

Page 6 Lines 189

“Figure 1” → “Figure 4”

Page 5, Lines 185-186

The sentence has been transferred into the description of hepatic function enzymes

“AST and ALT activities tended to increase in the mice treated with Vit D3,”

  1. Please mention the institutional animal care and use committee of Yamaguchi University (line 123-129) to 1 Animals.

Page 2, Lines 90-96

The sentences have been moved to 1 Animals as suggested.

“All procedures involving animals were approved by the Institutional Animal Care and Use Committee of Yamaguchi University and followed the Guidelines of Animal Care and Experiments of Yamaguchi University. The animal care and use program for Advanced Research Center for Laboratory Animal Science in Yamaguchi University has been accredited by AAALAC International since 2018. The Institutional Animal Care and Use Committee of Yamaguchi University approved this specific study under Approval No. 368.”

  1. Authors mentioned “Lungs” in line 216 and 272. Please show the figures of lungs or delete this word.

The word “Lungs” has been deleted as suggested.

Submission Date

20 September 2021

Date of this review

01 Oct 2021 17:01:12

We are most grateful that you are prepared to consider our manuscript for publication in biology and look forward to hearing from you at your earliest convenience.

Sincerely yours,

Dr. Tohru Kimura

Reviewer 2 Report

The authors of the paper ' Demonstrative experiment in the favorable effects of a static electric field in the treatment on Vitamin D3-induced hypercalcemia', demonstrated the benefits of SEF in preventing calcification of organs in mice treated with Vitamin D3.

Though the experiments done by the authors support their claim, there is some major information missing in the manuscript. They are as follows:

  1. The authors have not described the potential number of animals that were used in each set of the study. Also, does the data represent an average of experimental duplicates?
  2. Representing the statistical data using individual values than bar diagrams helps the reader to view the distribution range of each subject in that particular assay.
  3. Line 89- Does the highest dose mean 2.5 µg/kg?
  4. How do the authors determine the voltage of the SEF? Were there previous studies on this? Reference to those studies will be helpful to the readers.
  5. Were the hypercalcemia levels checked in the Vitamin D3 treated mice before they were subjected to SEF? If yes, then what method or supplementary data supporting the information should be provided in the manuscript.
  6. Figure 4 has a typing error and is referred to as Figure 1.
  7. Line 176, states the result which is not significant.
  8. The histopathology data should also have an image for the control mice.
  9. Hypercalcemia is supposed to affect the CNS, did the authors see any difference in these animals when treated with SEF.

Author Response

Comments and Suggestions for Authors

The authors of the paper ' Demonstrative experiment in the favorable effects of a static electric field in the treatment on Vitamin D3-induced hypercalcemia', demonstrated the benefits of SEF in preventing calcification of organs in mice treated with Vitamin D3.

Though the experiments done by the authors support their claim, there is some major information missing in the manuscript. They are as follows:

I am grateful to review #2 for the critical comments and useful suggestions that have helped me to improve our paper. As indicated in the response that follow, I have taken all these comments and suggestions into account in the revised version of this paper.

  1. The authors have not described the potential number of animals that were used in each set of the study. Also, does the data represent an average of experimental duplicates?

Page 3, Lines 98-99

The sentences have been corrected.

“Mice were assigned randomly to the following three groups: 1. Vit D3 treated mice (n = 5), 2. Vit D3 and SEF (Vit D3 + SEF) treated mice (n = 5) and 3. Untreated mice (n = 5).”

  1. Representing the statistical data using individual values than bar diagrams helps the reader to view the distribution range of each subject in that particular assay.

We submitted the raw data to the supplementary file.

  1. Line 89- Does the highest dose mean 2.5 µg/kg?

Page 3, Line 106

The sentence has been corrected.

“the dose of Vit D3 leaving all mice alive for 5 weeks was up to 0.25 μg/kg [11].”

  1. How do the authors determine the voltage of the SEF? Were there previous studies on this? Reference to those studies will be helpful to the readers.

We could not determine the voltage of the SEF and we try to determine the voltage with the following analyzer.

Recent study states the method for measuring power frequency electric exposure. An electromagnetic radiation analyzer (SEM-600, Beijing Safety Test Technology Co., Ltd., China) was used to measure the exposure intensity of power frequency electric field.

(Di G, Dong L, Xie Z, Xu Y, Xiang J. Effects of power frequency electric exposure on kidney. Ectoxicol Environ Saf. 2020; 194: 110354.)

  1. Were the hypercalcemia levels checked in the Vitamin D3 treated mice before they were subjected to SEF? If yes, then what method or supplementary data supporting the information should be provided in the manuscript.

Unfortunately, we did not examine Ca levels before the mice were subjected to SEF.

  1. Figure 4 has a typing error and is referred to as Figure 1.

We corrected our mistakes.

Page 6 Line 189

“Figure 1” → “Figure 4”

  1. Line 176, states the result which is not significant.

The sentences have been revised as suggested.

Page 5, Lines 185-187

“Although AST and ALT activities tended to increase in the mice treated with Vit D3, the Vit D3 treated group did not show significant differences as compared with the Vit D3 + SEF treated group and the untreated group.”

  1. The histopathology data should also have an image for the control mice.

We have added new microscopic photographs (C. Untreated group) in Figures 5, 6 and 7.

  1. Hypercalcemia is supposed to affect the CNS, did the authors see any difference in these animals when treated with SEF.

We could not find apparent CNS abnormality in the mice during this study.

We noted another abnormality in the Vit D3 treated mice and the sentence has been added.

Page 9 Lines 288-290

“In this examination, although echocardiograms could not be recorded in the mice, the Vit D3 treated group seemed to be slow in their behavior, suggesting the changes in cardiac legions and function.”

Submission Date

20 September 2021

Date of this review

11 Oct 2021 03:37:20

We are most grateful that you are prepared to consider our manuscript for publication in biology and look forward to hearing from you at your earliest convenience.

Sincerely yours,

Dr. Tohru Kimura

Round 2

Reviewer 1 Report

Authors replied correctly and sincerely. The reviewer's concerns were resolved.